# Quantitative analysis of protein-protein interactions and post-translational modifications in rare immune populations

Ayelet Avin[1], Maayan Levy[1], Ziv Porat[2] & Jakub Abramson[1]

In spite of recent advances in proteomics, quantitative analyses of protein–protein interactions (PPIs) or post-translational modifications (PTMs) in rare cell populations remain challenging. This is in particular true for analyses of rare immune and/or stem cell populations that are directly isolated from humans or animal models, and which are often characterized by multiple surface markers. To overcome these limitations, here we have developed proximity ligation imaging cytometry (PLIC), a protocol for proteomic analysis of rare cells. Specifically, by employing PLIC on medullary thymic epithelial cells (mTECs), which serve as a paradigm for a rare immune population, we demonstrate that PLIC overcomes the inherent limitations of conventional proteomic approaches and enables a high-resolution detection and quantification of PPIs and PTMs at a single cell level.

[1] Department of Immunology, Weizmann Institute of Science, Rehovot, 76100, Israel. [2] Department of Biological Services, Weizmann Institute of Science, Rehovot, 76100, Israel. Correspondence and requests for materials should be addressed to J.A. (email: jakub.abramson@weizmann.ac.il)

The past two decades have seen a dramatic progress in the development and optimization of both novel and traditional proteomics methods, including mass spectrometry, protein microarrays, proximity-based assays, and others[1–3]. This, in turn, has significantly advanced our understanding of molecular mechanisms underlying various biological and biochemical processes. However, in spite of this progress, the available proteomic approaches are mainly optimized for analyses of abundant cell populations, typically requiring millions of cells per analysis. This consequently poses serious limitations for similar types of analyses in rare cell populations that are directly isolated from humans or animal models. Often, the common approach to overcome these limitations is to substitute such primary cells with cell lines, which can be easily expanded and manipulated in vitro. However, such approaches may give rise to multiple artifacts, as the proteome compositions, protein–protein interactions (PPIs) and/or protein post-translational modifications (PTMs) in these models may substantially differ from those found in the corresponding primary cells.

In particular, quantitative analysis of PPIs or PTMs in various populations of the immune system remains technically very challenging, due to their rarity and/or due to the constraints of using multiple surface markers for their identification. A very good example of immune cells that are very difficult to analyze on a proteomic level is a rare population of the thymic stroma—the medullary thymic epithelial cells (mTECs). Although mTECs constitute <0.1% of cells in the thymus, they are essential for the establishment of immunological self-tolerance by facilitating both negative selection of self-reactive T cells[4,5] and agonist selection of Foxp3+ T regulatory cells[6,7]. Crucial to the key role of mTECs in purging self-reactive T cells, is their unique capacity to promiscuously express and present almost all self-antigens, including thousands of tissue-restricted antigen (TRA) genes, such as insulin[8]. Importantly, the promiscuous expression of such TRA genes in the thymus was shown to be mainly mediated by a single factor—the autoimmune regulator (Aire)[9]. In spite of the recent progress in our understanding of how Aire regulates expression of its target genes (reviewed in Anderson and Su[10]), most of the molecular insights into its *modus operandi* come from in vitro proteomic studies[11–16]. Validation of these findings under physiological conditions, including analyses of Aire's interacting partners or PTMs in bona fide mTECs, using currently available proteomic approaches remains technically not feasible.

To overcome these limitations, we sought to develop an analytical approach, which would enable a quantitative analysis of PPIs and/or PTMs in rare cell populations that are often defined by the expression of multiple surface markers, such as mTECs, in an accurate, quantitative, and reproducible manner. To this end, we developed Proximity Ligation Imaging Cytometry (PLIC), a new protocol, which exploits and combines advantages of proximity ligation assay (PLA) and imaging flow cytometry (IFC). Importantly, although our study uses mTECs as an experimental paradigm, we also demonstrate that PLIC is suitable for proteomic analysis of other populations of the immune system for which standard proteomic approaches have been technically challenging.

## Results

### The development of PLIC. 
PLA is a relatively recently established assay[17–19] suitable for studying PPIs and PTMs in cell cultures or tissue sections immobilized on glass cover slips with high specificity. Specifically, the PLA methodology utilizes a pair of oligonucleotide-labeled antibodies binding in close proximity (maximal 30–40 nm apart) to primary antibodies recognizing the targeted protein(s). When the oligo probes are in close proximity,

they support ligation of additional DNA strands (connector oligonucleotides) to create a DNA circle that subsequently templates a localized rolling circle amplification (RCA), during which a repeated sequence product is generated. Detection is achieved by the addition of complementary fluorescently labeled oligonucleotides[20]. This allows amplification of the resulting fluorescent signal by up to three orders of magnitude[21] and thereby enabling its efficient detection by fluorescence microscopy. Although the PLA assay enables a highly sensitive and robust analysis of protein associations, one of its key limitations is that it is much less suitable for proteomic analysis of rare cell populations and/or populations defined by expression of multiple surface markers, which are the typical hallmarks of many immune cell subsets.

To overcome these limitations, we sought to modify and potentiate the conventional PLA protocol to enable multi-parametric fluorescent analysis of single cells in suspension, in a quantitative manner. To this end, we coupled the PLA assay with IFC, which allows multiparametric fluorescent and morphological analysis of thousands of cellular events along with a statistical analysis of subcellular distribution of the measured signal in single cells[22,23] (Fig. 1a). Based on these combined features, the newly designed PLIC protocol seems ideal for quantitative and highly sensitive detection of PPIs and PTMs in rare immune cell subsets that are characterized by multiple surface markers.

**PLIC analysis of PPI in rare immune cells.** First, to verify that PLIC is indeed suitable for a quantitative analysis of PPIs in rare cell populations, we utilized mTECs as an experimental paradigm of a rare immune population and analyzed the interaction between Aire and its recently identified molecular partner—Sirtuin-1 (Sirt1)[24](Fig. 1b). Indeed, applying the PLIC protocol on CD80hi mTECs (mTEChi), directly isolated from a mouse thymus, demonstrated a clear PLA signal that was localized to nuclear speckles in ~ 25% of mTEChi cells, suggesting that at least in half of the Aire-expressing mTECs, Aire and Sirt1 are present in a very close mutual proximity. Importantly, no such signal was detected in thymic CD45+ populations, which served as a negative control, highlighting the specificity of the PLIC protocol (Fig. 1c, d, Supplementary Fig. 1a). Moreover, Aire and Sirt1 PLA signal co-localized (though not completely overlapped) with Aire staining (Fig. 1e, Supplementary Fig. 1b), suggesting that only a fraction of Aire protein interacts with Sirt1 at a given moment in a cell. Although Aire and Sirt1 interaction can also be detected by standard PLA protocol on thymus sections, acquired with confocal microscopy (Supplementary Fig. 2), the signal cannot be accurately quantified and linked to a specific cell population, defined by multiple molecular markers. As a result, conventional PLA protocol cannot distinguish a true signal from a false-positive signal caused by cells' auto-fluorescence or non-specific binding of PLA probes (Supplementary Fig. 3). In addition, it should be noted that tissue sections can be extremely heterogenous, which can lead to over- or under-representation of a specific cell population in a given tissue section, especially when such a population is scarce (Supplementary Fig. 2).

In contrast, PLIC utilizes advanced data processing algorithms, which allow to efficiently filter out possible false-positive signals. This together with multiparametric fluorescent and statistical analysis of PLA signal intensity and its subcellular distribution in thousands of single cells dramatically increases the accuracy, specificity, robustness, and reproducibility of the obtained data (Fig. 1d). Moreover, combining PLIC (assessing two interacting partners) with an overlapping staining for one of the partners or a third protein suspected to be in their proximity, may provide additional information about the actual fraction of proteins that

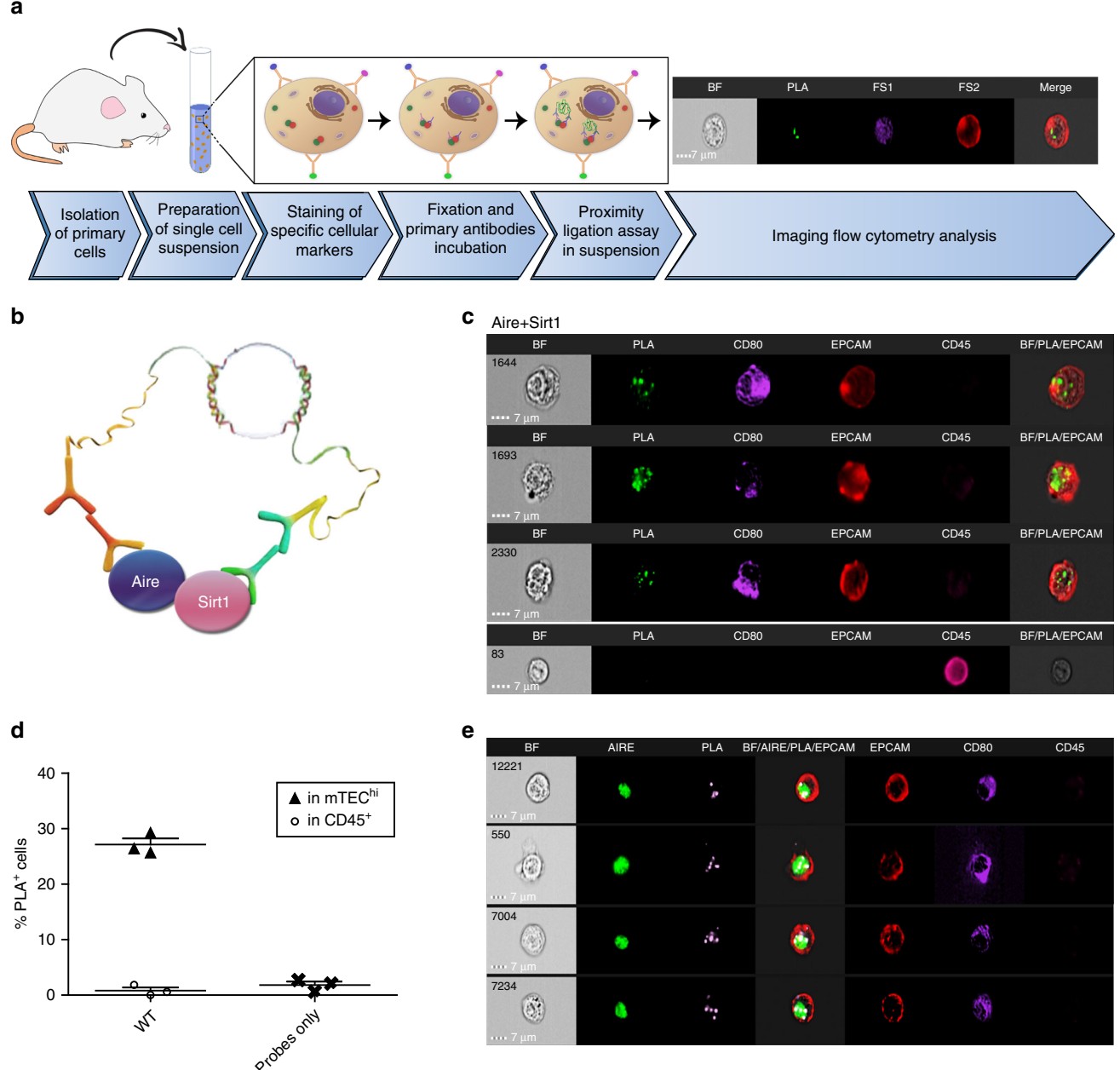

**Fig. 1** Detection and quantification of PPIs at a single cell level by PLIC. **a** A graphical scheme illustrating the main steps of the PLIC protocol. FS fluorescent signal. **b** A graphical scheme illustrating PLIC analysis of Aire and Sirt1 interaction. **c** PLIC analysis showing representative images obtained from imaging flow cytometry of the interaction of Aire and Sirt1 in mTEC$^{hi}$ and CD45$^+$ populations presented as (left to right): bright field (BF), PLA signal (green), staining of CD80 (PB), EpCAM (APC), CD45 (APC/Cy7), and BF/PLA/EpCAM overlay (merged). **d** Quantitative analysis of Aire and Sirt1 interaction (data are presented as percentage of positive cells (PLA$^+$) in mTEC$^{hi}$ or CD45$^+$ cells isolated from WT mice, compared to probes background. Shown are values from three independent experiments. Each point on the graph (triangle for mTEC$^{hi}$, circle for CD45$^+$, and cross for probes only) represents an averaged value of single independent experiment, calculated from 2 to 4 biological replicates, +s.e.m). **e** PLIC analysis showing representative images of the interaction of Aire and Sirt1 in mTEC$^{hi}$ overlaid with Aire staining (left to right): bright field (BF), Aire-AF488 (5H12), PLA signal (FarRed), BF/Aire/PLA/EpCAM overlay (merged), staining of EpCAM (APC/Cy7), CD80 (PB), and CD45 (PE/Cy7). Scale bar 7 μM

interact in a given time, or may help identifying additional proteins that may be in their proximity. Therefore, PLIC seems to be an ideal protocol for quantitative analysis of protein–protein interactions in rare immune populations.

**PLIC analysis of protein oligomerization in rare immune cells.** Next, to test whether PLIC can also be used for a quantitative analysis of homo-dimerization or oligomerization of proteins in

rare cell subsets, we analyzed the proximity of two Aire proteins in mTEC$^{hi}$ using a modified PLIC protocol (Fig. 2a). Although Aire has been previously shown to form homo-dimers and tetramers under in-vitro conditions[16,25–27], no assay has been able to validate the formation of such complexes under physiological conditions in bona fide mTECs. This is not only due to mTECs rarity, but also due to the hardships of isolating Aire from the nuclear matrix. Indeed, by applying the PLIC protocol, we were able to demonstrate for the first time that Aire undergoes

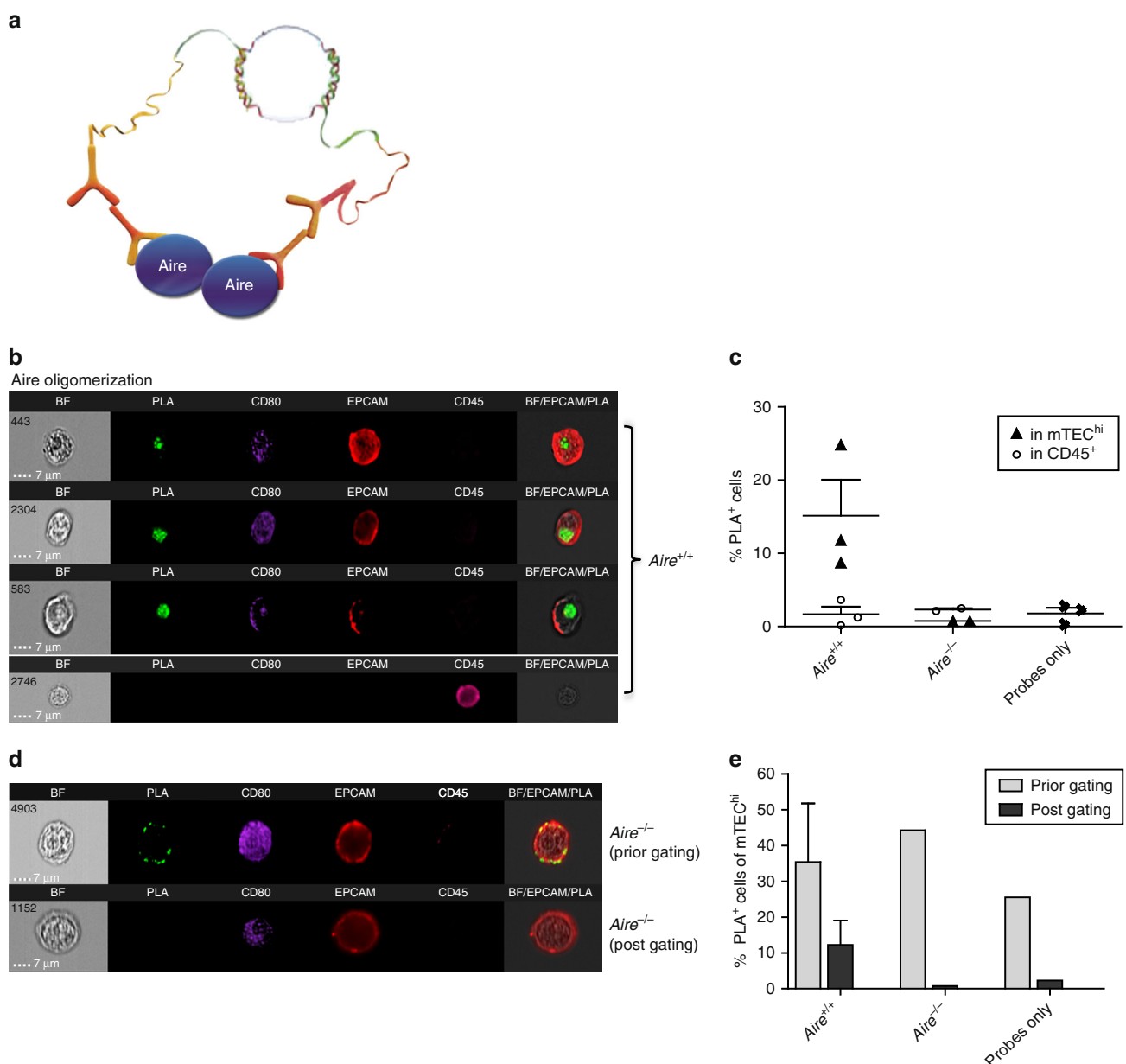

**Fig. 2** PLIC quantitative analysis of protein oligomerization in mTECs. **a** A graphical scheme illustrating PLIC analysis of Aire oligomerization using a single monoclonal antibody. **b** PLIC analysis showing representative images of the oligomerization of Aire in mTEC$^{hi}$ and CD45$^+$ populations, presented as (left to right): bright field (BF), PLA (green), staining of CD80 (PB), EpCAM (APC), CD45 (APC/Cy7), and BF/PLA/EpCAM overlay (merged). **c** Quantitative analysis of Aire oligomerization (presented as the percentage of PLA$^+$ cells) in mTEC$^{hi}$ or CD45$^+$ cells isolated from Aire$^{+/+}$ and Aire$^{-/-}$ mice, compared to probes background. Shown are values from three independent experiments. Each point on the graph (triangle for mTEC$^{hi}$, circle for CD45$^+$, cross for probes only) represents an average value of single independent experiment, calculated from 2 to 4 biological replicates, +s.e.m. **d** Representative images of Aire oligomerization as detected by PLIC in mTEC$^{hi}$ isolated from Aire$^{-/-}$ mice, prior to corrected analysis gating and post gating (filtering out false-positive signal). Images are presented as (left to right): bright field (BF), PLA (green), staining of CD80 (PB), EpCAM (APC), CD45 (APC/Cy7), and BF/PLA/EpCAM overlay (merged). Scale bar 7 μM. **e** Quantitative analysis of Aire oligomerization (presented as the percentage of PLA$^+$ cells) in mTEC$^{hi}$ cells isolated from Aire$^{+/+}$ (n = 2, s.e.m) or Aire$^{-/-}$ (n = 1) mice, compared to probes background sample (n = 1), prior to (light grey) and post (dark grey) corrected gating

homo-oligomerization under physiological conditions (Fig. 2b, c). Specifically, our data demonstrate that Aire homo-oligomers (detected as PLA signal in nuclear speckles) are detectable in ~ 10–25% of mTEC$^{hi}$ population, while they are virtually undetectable in thymic CD45$^+$ population, which served as a negative control (Fig. 2c). Such relatively high signal variability among analyzed samples was consistent in all independent PLIC analyses, possibly suggesting that Aire oligomerziation is a rather dynamic process.

Similarly as in the previous case, thorough analysis of subcellular signal distribution of recorded IFC images, enabled us to distinguish true PLA signal from non-specific background staining of the PLA probes and/or from cells' auto-fluorescence (Fig. 2b, d; Supplementary Fig. 3). Specifically, this was achieved by focusing only on nuclear and speckled PLA signal and by gating out either non-specific PLA signal, which is mainly localized to the plasma membrane, or by filtering out auto-fluorescent cells, which typically show a diffused fluorescent

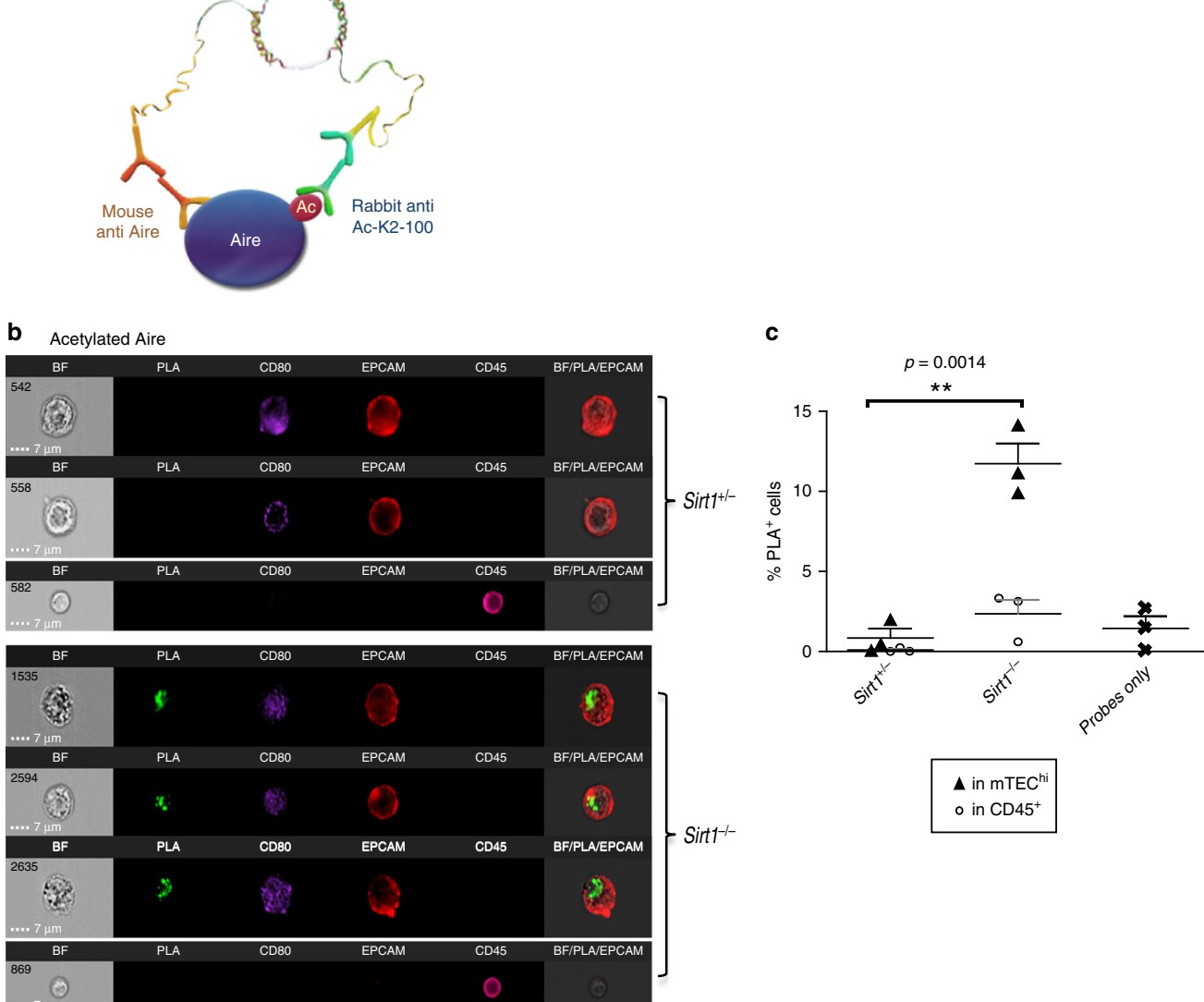

**Fig. 3** PLIC quantitative analysis of protein post-translational modification in mTECs. **a** A graphical scheme illustrating PLIC analysis of acetylated Aire, using mouse anti-Aire #209 antibody, and rabbit anti-lysine acetylation antibody. **b** PLIC analysis showing representative images of Aire acetylation in mTEC[hi] and CD45[+] cells isolated from Sirt1[+/−] and Sirt1[−/−] mice. Presented as (left to right): bright field (BF), PLA (green), staining of CD80 (PB), EpCAM (APC), CD45 (APC/Cy7), and BF/PLA/EpCAM overlay (merged). Scale bar 7 μM. **c** Quantitative analysis of Aire acetylation (presented as the percentage of PLA[+] cells) in mTEC[hi] or CD45[+] cells isolated from Sirt1[+/−] and Sirt1[−/−] mice. Shown are values from three independent experiments. Each point on the graph (triangle for mTEC[hi], circle for CD45[+], and cross for probes only) represents an average value of single independent experiment, calculated from 2 to 4 biological replicates, +s.e.m; **p-value = 0.0014 (independent student t-test)

signal (Supplementary Figs. 3 and 4). This allowed us to dramatically reduce the percentage of false-positive interactions (Fig. 2e, Supplementary Fig. 4), which would have been otherwise detected by conventional PLA protocol or PLA coupled to conventional flow cytometry[28–30]. This was further highlighted by similar PLIC analysis of Aire-deficient mTEC[hi] (Fig. 2d, e, Supplementary Fig. 3) in which all positive signal was membranal rather than nuclear. These data collectively demonstrate that PLIC analysis is not only quantitative, but also much more accurate than PLA coupled to conventional flow cytometry as it enables to filter out false signal based on its subcellular distribution.

**PLIC analysis of PTMs in rare immune cells**. We next sought to validate whether PLIC can also be applied for a quantitative

analysis of PTMs at a single cell level, again utilizing mTECs as an experimental paradigm. Specifically, we sought to analyze the levels of acetylated Aire in bona fide mTECs. As mentioned above, we have previously demonstrated that Sirt1 is an important partner of Aire, which is required for Aire-mediated promiscuous gene expression in mTECs[31]. Although we have previously shown that Sirt1 is able to deacetylate Aire on multiple lysine residues in transfected HEK-293 cells[24], experimental evidence validating that Aire is similarly deacetylated by Sirt1 in mTECs is still missing, due to the inherent limitations of conventional proteomic tools for analysis of rare cell populations. To better address this important question, we utilized a modified PLIC protocol, which we designed for quantitative analysis of PTMs, such as lysine acetylation of Aire, at a single cell level. To this end, we applied rabbit lysine acetylation-specific antibody (Cell Signaling #9814S; clone Ac-K2–100), along with Aire-

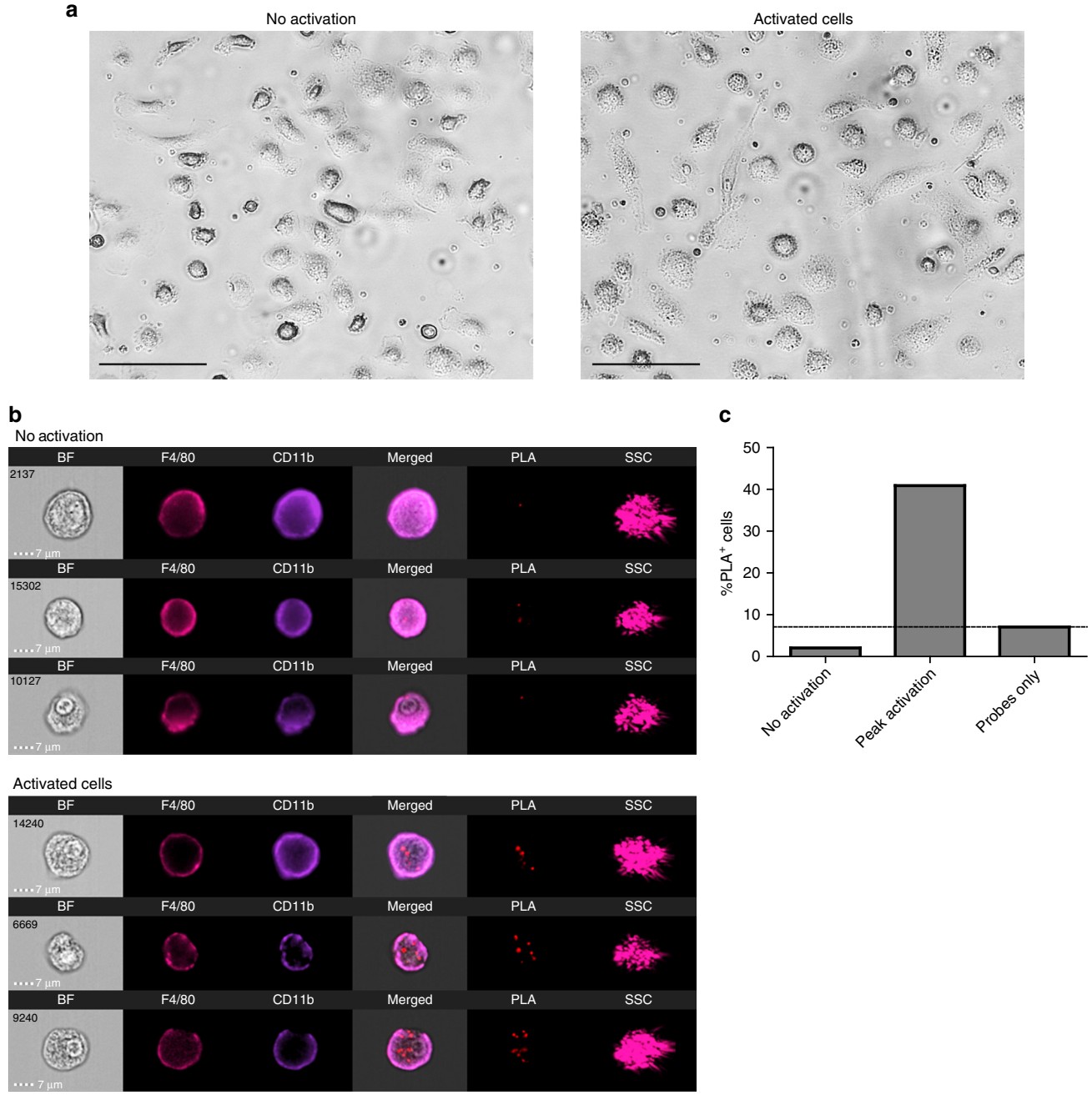

**Fig. 4** PLIC analysis of caspase-1 oligomerization in NLRP3 inflammasome-activated bone-marrow derived macrophages (BMDM). **a** Light microscopy images (x20) of non-activated BMDM and NLRP3 inflammasome-activated (LPS + ATP) BMDM. Scale bar 100 μM **b** Representative images of non-activated and NLRP3 inflammasome-activated BMDM, detected by PLIC, depicting median cells of the PLA bright detail intensity, and the PLA max pixel features. Cells are presented as (left to right): bright field (BF), staining of F4/80 (PE/Cy7), CD11b (PB), BF/PLA/F4/80/CD11b overlay (merged), PLA (FarRed), and side scatter (SSC). Scale bar 7 μM. **c** Quantitative analysis of caspase-1 oligomerization (presented as the percentage of PLA+ cells) in non-activated cells, cells at peak activation, and probes only control. Representative of four independent experiments with similar results, calculated from 20,000 cells (F4/80+CD11b+) per treatment (no activation/ NLRP3 inflammasome activation by LPS + ATP)

specific mouse monoclonal antibody (mAb) (generated de novo in the lab) in order to detect acetylated Aire by PLIC (Fig. 3a). Strikingly, while we were not able to detect any true positive signal in mTEC$^{hi}$ isolated from Sirt1$^{+/-}$ mice, we detected a clear positive signal in mTEC$^{hi}$ isolated from Sirt1$^{-/-}$ mice (Fig. 3b, c). These data, therefore, validate that Sirt1 is indeed required for effective deacetylation of Aire in-vivo, and suggest that this modification is tightly regulated and may serve as a molecular switch for Aire activation. Once more, PLIC analysis emerged to be highly specific and accurate, and to our knowledge represents

the only available method capable of detecting and quantifying PTMs in rare cell populations like mTECs, without the need for an antibody specific to the given post-translationally modified protein.

Moreover, PLIC analysis of Aire acetylation also highlighted one of the key limitations of using signal co-localization in IFC (i.e., without the combination with PLA), as a possible measure of physical proximity between two protein targets. This type of analysis may be highly inaccurate since the mere observation of overlaid pixels, where the two proteins are probed, can be a

consequence of random juxtapositioning[32] (resulting from the machine resolution limitations), especially when the signal is diffused or the protein is highly abundant in the cell. Indeed, this is well illustrated by the high signal overlap between Aire and acetylated lysines, imaged by IFC alone without the use of PLA (i.e., probing for Aire and general Lysine-Ac with regular fluorescent-conjugated secondary antibodies) (Supplementary Fig. 5a). Measuring the two proteins proximity by such co-localization can consequently lead to a dramatic overestimation of a true (i.e., non-random) interaction (Fig. 3c, Supplementary Fig. 5b), and thus to erroneous conclusions.

**PLIC can be used universally**. Finally, in order to demonstrate that PLIC can be applied universally to study PPIs and PTMs in immune cells and proteins other than mTECs and Aire respectively, we next utilized it for analysis of caspase-1 (Casp1) oligomerization[33] in macrophages following NLRP3 inflammasome activation. Inflammasomes are multiprotein signaling platforms of the innate immune system, which become activated in response to "danger" signals (e.g., infectious microbes and/or "danger" molecules derived from host proteins)[34-36]. Once active, the inflammasomes mediate pro-inflammatory responses via activation of the Casp1-dependent cascade. Common methodologies currently used to detect NLRP3 inflammasome activation are Western blot analysis of Casp1 cleavage, downstream cytokines activation[37-39], microscopic analysis of transfected cells with reporter fused inflammasome related proteins[40,41], and mouse knockout models[42-44]. However, none of these methods fully embodies the actual physiological environment in which these processes occur, nor enables analysis of more specifically defined, rare populations in which the inflammasome is activated. Therefore, to test whether NLRP3 inflammasome activation can be quantitatively analyzed by PLIC under more physiological settings, we analyzed endogenous Casp1 oligomerization in mouse bone marrow-derived macrophages (BMDM) in response to dual stimulation by lipopolysaccharide (LPS) and adenosin triphosphate (ATP). Indeed, using the PLIC protocol, we were able to detect and quantify, to the best of our knowledge and for the first time, Casp1 oligomerization (as a measure of NLRP3 activation) in a non-transgenic system, thus validating that PLIC can be used universally in other biological systems (Fig. 4). In this specific manner, PLIC can be very instrumental for future studies of inflammasome activation in different physiological and pathological contexts.

## Discussion

Over the past decade, genomic research has experienced substantial progress thanks to the development of various highly innovative technologies, such as Next Generation sequencing, which now allow in-depth and/or high-throughput genomic analyses at a single-cell level. In contrast, however, the progress in proteomic research has been considerably slower, especially with regards to the sensitivity and specificity of the currently available methods. This consequently results in significant challenges in quantitative proteomic analyses of rare cell populations that are directly isolated from humans or animal models, and which are often characterized by multiple surface markers. In this study we tried to overcome these limitations by developing a novel proteomic tool—PLIC—enabling a quantitative, robust, and accurate analysis of PPIs and PTMs at a single-cell level (Fig. 5).

To validate that PLIC is indeed suitable for such type of analyses, we utilized mTECs as an example of a rare immune population, which represents a very challenging target for various types of proteomic analyses, including quantitative analysis of PPIs and PTMs. Indeed, by utilizing the PLIC protocol, we were

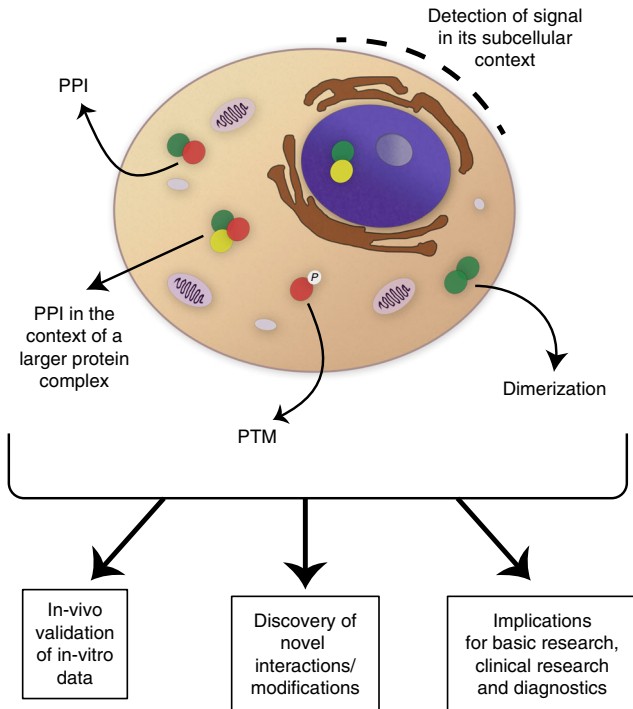

**Fig. 5** Graphical scheme illustrating the various applications of PLIC. PPI protein–protein interaction, PTM post-translational modification

able to obtain several in-vivo biological findings, which provide important molecular insights into the mechanisms underlying the establishment of central immunological tolerance in the thymus, and which have remained unachievable by currently availably proteomic tools. Specifically, we demonstrate, to the best of our knowledge and for the first time, that Aire undergoes homo-oligomerization under physiological conditions in a large fraction of mTECs that were directly isolated from a mouse thymus. This finding is of particular importance, as it helps better elucidating and validating the pathogenesis of recently identified mono-allelic mutations in the *AIRE* gene, which were found to cause common organ-specific autoimmune diseases in a dominant negative manner[27]. Furthermore, although Aire has been shown to physically interact with over 20 different proteins (reviewed in Abramson and Goldfarb[45]), these interactions were thus far validated only under highly non-physiological conditions, including transfected cell lines, far-Western Blot analysis, or yeast 2-hybrid systems. By analyzing the interaction between Aire and its previously reported partner Sirt1[24] in bona fide mTECs, we demonstrate that PLIC is also well suited for quantitative analysis of PPIs in rare immune populations. Importantly, we also provide experimental evidence that PLIC offers a very powerful tool for quantitative analysis of PTMs of proteins at a single cell level. Specifically, we demonstrate, for the first time, that Aire undergoes Sirt1-dependent deacetylation in mTECs, further elucidating and validating the in vivo significance of Sirt1 in the establishment of immunological self-tolerance. Finally, by utilizing PLIC on additional experimental paradigms, such as the quantification of NLRP3 activation in BMDMs, we demonstrate that PLIC can be used universally for proteomic analyses of any rare population of choice, to which a flow cytometry grade antibody is available. For instance, PLIC could be very instrumental for quantitative proteomic analyses in various rare immune and/or stem cell populations. These may in particular include analyses of oligomerization of various immunoreceptors (e.g., antigen, cytokine, or Toll-like receptors), as well as analyses of downstream

signaling and/or transcriptional events, which could be determined by analyzing PPIs and/or PTMs of specific signaling molecules and/or transcriptional regulators, respectively. Importantly, PLIC can be used not only for in-vivo validation and quantification of PTMs and PPIs, but also for de novo discovery (Fig. 5).

In recent years, a number of different proximity-based techniques for studying protein–protein interactions have emerged (reviewed in Peter Lönn and Ulf Landegren[3]). In addition to PLA, these assays particularly include (i) protein fragment complementation assays (PCA), (ii) proximity-dependent biotin labeling methods (using biotinylase-based biotin identification (BioID) or ascorbate peroxidase (APEX)), and (iii) Förster resonance energy transfer (FRET). However, all of the above methods share several inherent limitations. First, many of these methods (e.g., PCA, BioID, or APEX) require genetic manipulation or overexpression of the targeted protein(s), and thus do not allow analysis of endogenous proteins. Correspondingly, neither of these methods is suitable for quantitative proteomic analysis of rare cell populations, directly isolated from humans or animal models. As we have demonstrated in this study, PLIC overcomes the key limitations of these currently available methods and enables highly sensitive and quantitative analyses of rare and complex cell populations. Specifically, the key advantages provided by PLIC include quantitative and multiparametric fluorescent analysis of thousands of single cells, statistical analysis of signal intensity and its subcellular distribution, as well as high specificity and reproducibility.

For instance, PLIC offers several important advantages over FRET-based approaches, which, like PLIC, are also suitable for assessing proximity between two endogenous proteins. First, due to high signal amplification, PLIC (and other PLA-based methods) have a much higher signal-to-noise ratio than FRET, which consequently dramatically improves detection sensitivity[3]. Second, PLIC offers a much broader and more universal choice of fluorescent probes than FRET, in which the repertoire of the donor and acceptor fluorescent pairs is very limited to comply with minimal spectra overlap required for efficient energy transfer[46,47]. As a result, PLIC can exploit more fluorescent channels for analysis, and thus allow detection of additional surface markers. A possible disadvantage of PLIC in comparison to FRET is that FRET signal exhibits a strict linear correlation with the expression levels of proteins, while PLA signals reach saturation at high expression levels[48], and thus have a limited dynamic range of detection.

Furthermore, PLIC overcomes many key limitations of the currently available PLA-based methods. Although PLA has been used in combination with either confocal microscopy or conventional flow cytometry (PLA-FACS)[28–30], the combination of PLA with IFC in PLIC enables a far more accurate, broader, and more sensitive detection of PPI or PTM than these previously described protocols. Specifically, as already discussed before, the key limitation of PLA followed by confocal microscopy is that it does not allow quantitative multi-parameter analysis of rare cells that are defined by multiple markers. Moreover, analysis of the fluorescent signal on tissue sections may be very inaccurate and problematic, as a result of very high variation among individual slides, often leading to over- or under-representation of a specific cell population in a given tissue section. As demonstrated by previous[28–30] and our studies, both PLA-FACS and PLIC are suitable for multiparametric analysis. However, unlike PLA-FACS, PLIC is much more accurate as it allows a very efficient filtering of auto-fluorescence and/or non-specific signals (caused by non-specific binding of the conjugated antibodies), by utilizing advanced features of the IFC imaging analysis including signal shape and localization. Moreover, PLIC does not require

customized conjugation of PLA probes to primary antibodies (as done in PLA-FACS[28–30]), but rather utilizes secondary antibodies conjugated to the PLA probes, which are commercially available. This consequently makes PLIC a very attractive and cost-effective assay, which can be used universally for studying PPIs and PTMs of a very broad spectrum of targets at a single cell level. A possible disadvantage of PLIC in comparison to conventional PLA followed by confocal microscopy is that PLIC does not provide information about the spatial localization within the tissue. Therefore, if such information is needed, it is advisable to also perform classical PLA on tissue sections in addition to PLIC. Other than the basic requirements for samples to be suitable for flow cytometry analysis, and the availability of an Imaging flow cytometer, which is rapidly expanding across research facilities around the world, the primary limitation of PLIC relates to the relatively high variability between individual sample measurements. Even though PLIC is optimized for analysis of rare populations, given that it is a multi-step protocol, there is a considerable risk of losing cells during the many incubation and washing steps. In addition, seen as protein interactions and/or modifications are dynamic processes, and that the analysis of rare populations is by default more prone to variations in signal detection, the manner in which the interaction is amplified by an RCA reaction, is more likely to give rise to higher variances in signal detection among samples. However, these issues can be easily resolved by increasing the number of experimental and sample replicates (>3). Another important clarification, which should be made, is that differences in PLA signal (higher intensity and/or bigger area), does not necessarily correlate to more interactions. Such differences in signal could arise from variations in sample preparation (e.g., antibody binding efficiency, probes concentration per cell, polymerase efficiency in creating the RCA reaction, and so on). For this reason, we suggest that for a reliable analysis, one should quantify the percentage of positive cells in which a PLA signal has been detected (independent of size/ intensity, as long as it exceeds the background threshold), within a given population.

In summary, PLIC is to the best of our knowledge, the first protocol to provide a sensitive, quantitative, and reproducible detection of PPI and PTM in (a) extremely rare populations and/ or (b) populations defined by many surface markers, (c) all in a physiological context (i.e., without the requirement for reporter/ tagged transgenic mice). As such, PLIC is a powerful, yet technically feasible and affordable methodology, which significantly expands our proteomic toolbox for successful delineation of molecular mechanisms underlying diverse physiological and pathological processes, and which can dramatically accelerate proteomic research in multiple fields of biology, in particular immunology. Importantly, the PLIC protocol can be implemented not only to address basic biological questions, but may also be instrumental for diagnostics and clinical research (Fig. 5).

## Methods

**Mice.** All mice used in this study were maintained under specific pathogen-free conditions at the Weizmann Institute's animal facility and were handled in accordance to the guidelines of the Institutional Animal Care and Use Committee Number 33760217-2. Wild type C57BL/6 (B6) mice were purchased from Harlan Laboratories. The Sirt1$^{-/-}$ mutant mice[49] were kindly provided by M.W. McBurney and were maintained on a mixed 129sv/CD1 genetic background. B6.Aire$^{-/-}$mice were obtained from Jackson Laboratories.

**Thymic single cell preparation and primary staining.** Thymi were surgically removed from 3 to 5 week old mice and placed into cold PBSx1 supplemented with 2% Fetal Bovine Serum (FBS, Invitrogen). Thymi were trimmed of fat and connective tissues, and disintegrated by sequential enzymatic digestion using Collagenase D (Roche #1088858) and Collagenase-Dispase cocktail (Roche #269638). After 15 min of incubation, the fragments were pipetted up/down to aid the digestion mechanically, repeating every 10 min until reaching a single cell

suspension. The cells were filtered through a 50 micron mesh filter and washed with MACS buffer (PBSX1, 5 mM EDTA, 2% FBS) at 340 g for 5 min. Thymic stroma, containing the TEC fraction, was enriched using a Percoll gradient: the cells were suspended in 2 ml of 1.115 g/ml isotonic Percoll (Sigma #P1644), topped by 1 ml of isotonic 1.065 g/ml Percoll and 1 ml of PBSX1. Percoll gradient was centrifuged at 1467 g, 4 °C, with no breaks for 40 min. The thymic stroma accumulated between the top and middle layers and was collected and washed with MACS buffer at 340×g for 5 min. The isolated cells were stained in 200 μl of MACS buffer, with the specific fluorophore-labeled antibodies (1:200, Biolegend) for 40 min on ice: APC anti-mouse CD326 (EpCAM) (#118214; clone G8.8), APC/Cy7 anti-mouse CD326 (EpCAM) (#118218; clone G8.8), APC/Cy7 anti-mouse CD45 (#103115; clone 30-F11), PE/Cy7 anti-mouse CD45 (#103115; clone 30-F11), and Pacific Blue™ anti-mouse CD80 (PB, #104723; 16-10A1). Intracellular staining required further fixation steps in 80% cold acetone[50] (Sigma, diluted in DDW). Prior to fixation, each sample was suspended in 300 μl of 50% FBS-PBSx1, then 900 μl of 80% acetone was added to each sample, in three consecutive rounds of 300 μl, and the samples were incubated for 1 h at 8 °C. The cells were centrifuged for 10 min, 300 g, then washed with 10% FBS-PBSx1 and centrifuged again for 10 min, 300 g. Fixed cells were then incubated with primary antibodies for 1 h at RT, according to the relevant experiment—Acetylated Aire: anti-Aire #209 (1:100, Mouse monoclonal, self-made) + anti-pan-acetylated-lysine (1:100, Cell Signaling #9814 S; clone Ac-K2-100), Aire oligomerization: anti-Aire #209 (1:100, Mouse monoclonal, self-made), and Aire + Sirt1: anti-Aire #209 (1:100, Mouse monoclonal, self-made) + anti-Sirt1 (1:200, Millipore #07-131; rabbit polyclonal). The cells were then washed and incubated ON in 1 ml of 10% FBS-PBSx1 at 4 °C. Next day proceeded with PLIC protocol (see below).

**BMDM preparation and primary staining**. Femur and tibia bones were removed from 8 to 11 week old mice in a sterile hood. Bones were flushed with cold PBSx1 using a syringe. Cells were centrifuged at 340 g for 4 min, and pellets were treated with ACKx1 (2 ml) for 4 min, RT. The cells were washed with PBSx1 and centrifuged at 340 g for 4 min. The cells were re-suspended in medium (RPMI, 10% FBS, P/S) + MCSF (0.01 μg/ml) and seeded on 10 cm plates. The cells were incubated at 37 °C for a week. For NLRP3 inflammasome activation, BMDM were primed with LPS (1 μg/ml) for 2 h, and then activated with ATP (5 mM or 0 mM). The cells were collected with Trypsin C, blocked for FCγR with anti CD16/32 (1:100, Biolegend #101302) for 10 min, RT and stained in 200 μl (1:100) MACS buffer, with PB anti-mouse CD11b (Biolegend #101223) and PE/Cy7 anti-mouse F4/80 (Biolegend #123113) for 25 min at 4 °C. Intracellular staining required further fixation steps in 80% cold acetone (diluted in DDW, as described for thymic preparation). Fixed cells were further blocked with a general goat anti-mouse antibody (1:250, 1 h, RT), and then stained with primary antibody (mouse monoclonal anti caspase-1, AG20B0042-c100, 1:200), for 1 h, RT. The cells were then washed and incubated ON in 1 ml 10% FBS-PBSx1 at 4 °C. Next day proceeded with PLIC protocol (see below).

**PLIC protocol**. Samples containing single cell suspensions stained with fluorescently labeled surface antibodies and non-conjugated primary antibodies (as described above) were next incubated with Duolink® PLA probes, which were added according to manufacture guidelines (1:5 from Stock solution, diluted in 10% FBS-PBSx1). 50 μl of the probes mix was added per sample in the following combinations: for the detection of Acetylated Aire or Aire and Sirt1 interaction—PLA Rabbit PLUS and PLA Mouse MINUS proximity probes (Sigma #DUO92002, #DUO92004) and for the detection of Aire oligomerization and Casp1 oligomerization—PLA Mouse PLUS and PLA Mouse MINUS proximity probes (Sigma #DUO92001, #DUO92004). The samples were incubated in Eppendorf tubes for 1 h in 37 °C, gentle shake (300 RPM). After incubation, the samples were washed twice with 1 ml of Buffer A (0.01 M Tris, 0.15 M NaCl and 0.05% Tween 20, pH 7.4, filtered through 0.22 μm filter) + 10% FBS, then centrifuged for 12 min, 300 g, 4 °C. Ligation and amplification steps were done using Duolink® In Situ Detection Reagents Green (Sigma #DUO92014) or Duolink® In Situ Detection Reagents FarRed (Sigma # DUO92013). Ligation step was done according to manufacture guidelines (1:5 from ligation stock, 1:40 Ligase, diluted in UPW), in 50 μl of ligation mix per sample. The samples were incubated for 30 min in 37 °C, gentle shake (300 RPM). After incubation, the samples were washed twice with 1 ml of Buffer A + 10% FBS, then centrifuged for 12 min, 300 g, 4 °C. Amplification step was done according to manufacture guidelines (1:5 from Amplification stock, 1:80 Polymerase, diluted in UPW), in 50 μl of amplification mix per sample. The samples were incubated for 100 min in 37 °C, gentle shake (300 RPM), in the dark. After incubation, the samples were washed twice with 1 ml of Buffer B (0.2 M Tris and 0.1 M NaCl, pH 7.5, filtered through 0.22 μm filter) + 10% FBS, then centrifuged for 12 min, 300 g, 4 °C. Finally, the cells were re-suspended in 20–30 μl of 10% FBS-PBSx1, and acquired by multispectral imaging flow cytometry (ImageStreamX mark II flow cytometer; Amnis Corp., part of EMD—Millipore).

**PLIC—IFC overlay**. PLIC analysis aimed at Aire and Sirt1 interaction was done as described above followed by a second Aire probing, at the end of the PLIC protocol, by using a different monoclonal anti-Aire antibody conjugated to AF488 (1:100, eBioscience, #53–5934–82; clone 5H12), incubated for 1 h at RT.

**PLIC acquisition**. Approximately 1000 TECs (CD45⁻EpCAM⁺) and control cells (CD45⁺EpCAM⁻) were collected from each sample for TECs experiments. Approximately 20,000 BMDM (F4/80⁺CD11b⁺) were collected for inflammasome experiments. Data were analyzed using image analysis software (IDEAS 6.2; Amnis Corp) (see below). Images were compensated for fluorescent dye overlap using single-stain controls.

**PLIC data analysis**. In order to quantify PPI or PTMs, we measured the percentage of PLA⁺ cells in the relevant population. Specifically, we measured the percentage of PLA⁺ cells in a specific biological replicate (single mouse) and then averaged these biological replicates for each experimental group (e.g., WT vs. KO) per each individual experiment. The statistical significance was then determined by Student t-test using the averaged values obtained from each individual independent experiment.

**TEC analysis**. Cells were gated into "mTEC^hi" and "mTEC^low" based on the intensity of CD80 staining, and PLA⁺ cells were gated using the Bright Detail Intensity (the intensity of localized bright spots within the masked area in the image) and Max Pixel (the largest value of the background-subtracted pixels in the image) features of the PLA channel. The gate was set according to a sample containing only the PLA probes, without the primary antibody ("probes only"), which served as a background control. In order to further decrease non-specific signal resulting in high background (which typically shows membranal localization), we further gated the PLA⁺ cells according to the signal sub-cellular localization using the above features on the AdaptiveErode_78 mask (identifies pixels that will form a circle that touch the input boundary with at least a prescribed radius threshold. The result is an adaptive erosion that takes shape into account). Next, for analyses in which target proteins are expressed exclusively in the nucleus, we gated the cells based on the Max Contour Position feature (the location of the contour in the cell that has the highest intensity concentration mapped to a number between 0—object center and 1—object perimeter), and the area of the PLA staining, using the Morphology mask (includes all pixels within the outermost image contour) in order to minimize false-positive signal caused by non-specific binding or autofluoerscence. To further eliminate artifacts (e.g., penetrance of fluorescence from one channel to another, which cannot be resolved by compensations), we gated the cells based on their Bright Detail Similarity (BDS; compares the small bright image detail of two images and can be used to quantify the co-localization of two probes in a defined region—the log transformed Pearson's correlation coefficient of the localized bright spots with a radius of 3 pixels or less within the masked area in the two input images), to exclude cells with BDS > 1.5 (i.e., reflecting strong signal overlay between channels).

**BMDM analysis**. PLA⁺ cells were gated using the Bright Detail Intensity and Max Pixel features of the PLA channel, as described above. The gate was set according to a sample containing only the PLA probes, without the primary antibody ("probes only").

**Data availability**. The data that support the findings of this study are available from the corresponding author upon reasonable request.

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

## Acknowledgements

We thank Dr. Eran Elinav for providing materials and technical support for the inflammasome experiments. The research in the Abramson lab is kindly supported by the by the European Research Council Consolidator Grant—ERC-2016-CoG-724821, Israel Science Foundation (722/14 and 1796/16); Sy Syms Foundation; Binational Science Foundation; Rising Tide Foundation; Maurice and Vivienne Wohl Charitable Foundation; Goodman Family Charitable Lead Annuity Trust; and Ruth and Samuel David Gameroff Family Foundation. J.A. is an incumbent of the Dr. Celia Zwillenberg-Fridman and Dr. Lutz Fridman Career Development Chair.

## Author contributions

J.A. and A.A. designed the study and wrote the manuscript; A.A. performed most of the experimental work; A.A. and Z.P. preformed and designed the IFC analysis. M.L. helped with BMDM samples preparations. M.L., Z.P. and J.A. helped in performing, analyzing, and/or designing some of the experiments.

## Additional information

**Competing interests:** The authors declare no competing financial interests.

