## [Peer Review File · Nature Communications]

Reviewers' Comments:

Reviewer #1:

Remarks to the Author:

The manuscript by Avin, et al. combined proximity ligation with multispectral imaging flow cytometry, developed a novel protocol for analyzing protein-protein interactions and even post-translational modifications in rare cell population or mixed cell population that require flow cytometry to analyze specific cell subset. The examples they present include oligomerization of Aire, Aire and Sirt1, acetylation of Aire, and Caspase 1 oligomerization in mTECs and BMDMs, respectively. The protocol overcomes the limitation of purified primary cell numbers required in Duolink and other common proteomic approaches. It provides a possibility to study protein-protein interaction and protein modification in mixed cell population. With some modification, it can be used in wider field. There are some problems with this protocol.

1. the expressions of CD80 in some samples look odd, Some explanations are needed. Without PLA treatment, do they find similar pattern of CD80 expression?
2. one of the strengths of this combined technique is to analyze hundreds or even thousands of primary cells in a mixed cell population. It is not clear whether the given quantitative data showed the results of this many cells. The figures mostly showed an average of 3 experiments instead of an average of 1,000 cells. More analytical data can be given if the cells show different patterns of Aire oligomerization and cell phenotype (including CD80 expression).
3. it will be more informative if the authors could compare their technique with confocal microscopy and other imaging technology.

Reviewer #2:

Remarks to the Author:

The manuscript entitled "Quantitative analysis of protein-protein interactions and posttranslational modifications in rare immune populations" by Avin et al. describes using of proximity ligation assay (PLA) with flow cytometry readout for analyses of proteins, protein-protein interactions and posttranslational modifications. The authors have used the assay to quantify the interaction of Aire and Sirtuin-1 in rare mTEC cell populations, and to illustrate Sirt-1 dependent deacetylation of Aire in mTEC.

Main concerns:

The main focus of the manuscript is to present a new technology based on flow cytometry readout combined with PLA. The authors have used a commercial PLA kit (Duolink), followed the protocol, and have performed the PLA with flow cytometry readout. The authors refer to some early publications with PLA based on microscopy imaging. However, they have completely missed the later publications, where the PLA has been combined with flow cytometry; Leuchowius et al. (Cytometry, 2009), is one of the earliest publications in which the PLA-flow cytometry was demonstrated. Löff et al. (Mol. Cellular Proteomics, 2017) have presented data on detection of BCR-ABL fusion proteins in blood samples from leukemia patients using PLA-flow cytometry. Löff et al. (Scientific reports, 2016) have also used a more sophisticated multicolor/multiplex PLA-flow cytometry for detection of exosomes. Also, Frei et al. (Nature Methods, 2016) have used PLA in CyTOF for multiplex detection of RNAs. The biological data presented in the manuscript are, as pointed out by the authors, not novel either. The only modest new data is the demonstration of "homo-oligomerization under physiological conditions".

Minor remarks:

On page 3, first paragraph: wrong description of PLA; the DNA oligonucleotides conjugated to antibodies are not ligated to the other oligonucleotides.

Figure 1a, The PLA using Duolink kit is also wrongly illustrated. And also not visible without

enlargement.

As all the technology and data presented here are previously known, my recommendation is to not try to publish this manuscript.

Referee #1

1. The expressions of CD80 in some samples look odd, Some explanations are needed. Without PLA treatment, do they find similar pattern of CD80 expression?

Re: Indeed, the CD80 staining in mTECs is not as distinct as that of other surface molecules such as MHCII or EpCAM, which in many (but not all cells) give very clear signal, characterized by increased intensity on the cells' surface, but could also have a rather diffused character (similar to that of CD80 staining). The reason for such a rather poor staining is likely the quality of the commercially available antibody and the CD80 protein expression levels on mTECs surface. We would like to stress however that even this staining is sufficient to distinguish between mTEC^{high} (higher signal intensity) and mTEC^o (no-low signal intensity) populations and correlates very well with MHC-II staining, which can be used as an additional/alternative mTEC maturation marker. CD80 staining looks the same whether we add the PLA process or not, as can be seen in these images acquired in a separate experiment of primary antibody calibrations (without PLA):

2. One of the strengths of this combined technique is to analyze hundreds or even thousands of primary cells in a mixed cell population. It is not clear whether the given quantitative data showed the results of this many cells. The figures mostly showed an average of 3 experiments instead of an average of 1,000 cells. More analytical data can be

given if the cells show different patterns of Aire oligomerization and cell phenotype (including CD80 expression).

Re: Although the information is in fact included in the text, the referee's comment suggests that it may not be entirely clear to the reader and that we should probably explain better how exactly our data are analyzed.

In the "experimental procedures" section of the main text we mention how many cells were acquired per each sample – "Approximately 1,000 TECs (CD45-EpCAM+) and control cells (CD45+EpCAM-) were collected from each sample for TECs experiments. Approximately 20,000 BMDM (F4/80+CD11b+) were collected for inflammasome experiments."

In the figure legend (per each figure), we address the number of experiments done, and number of animals in each experiment, for example – "Quantitative analysis of Aire and Sirt1 interaction (presented as the percentage of PLA⁺ cells) in mTEChi or CD45+ cells isolated from WT mice, compared to probes background (Average of 3 independent experiments, n=3)", meaning each point on the graph (triangle for mTEChi, circle for CD45+) represent a single independent experiment, showing the average of 2-4 animals which were used for that experiment.

Seen as we are interested in information about the mTEC population as a whole (and not as individual cells), data is presented as percentage of positive cells (PLA+) out of the mTEChi population, and as mentioned before each dot in the graph present the average of such calculation per all animals of the same group (WT/KO) in that individual experiment.

Although PLA signal can look differently between cells (intensity and area wise), it doesn't necessarily correlates to more interactions/ different interaction pattern. It could arise from technical differences between samples (i.e antibody binding efficiency, probes concentration per cell, polymerase efficiency in creating the rolling circle amplification), and therefore we felt that the most reliable analysis would be that of % of positive signal within the population.

We will be happy to be more specific regarding this point in the revised manuscript.

3. It will be more informative if the authors could compare their technique with confocal microscopy and other imaging technology.

Re: Indeed, this is a great point. Although in Supp.Fig2 we demonstrate the difference in detection of PLIC versus standard PLA done on thymus sections, we did not specify anywhere in the text that the images were taken with a confocal microscopy. Therefore we would like to thank referee #1 for bringing this problem to our attention. We will be happy to address it and further elaborate on advantages that PLIC offers in comparison to classical fluorescence microscopy.

Referee #2

1. The main focus of the manuscript is to present a new technology based on flow cytometry readout combined with PLA. The authors have used a commercial PLA kit (Duolink), followed the protocol, and have performed the PLA with flow cytometry readout. The authors refer to some early publications with PLA based on microscopy imaging. However, they have completely missed the later publications, where the PLA has been combined with flow cytometry; Leuchowius et al. (Cytometry, 2009), is one of the earliest publications in

which the PLA-flow cytometry was demonstrated. Löff et al. (Mol. Cellular Proteomics, 2017) have presented data on detection of BCR-ABL fusion proteins in blood samples from leukemia patients using PLA-flow cytometry. Löff et al. (Scientific reports, 2016) have also used a more sophisticated multicolor/multiplex PLA-flow cytometry for detection of exosomes. Also, Frei et al. (Nature Methods, 2016) have used PLA in CyTOF for multiplex detection of RNAs.

Re: For some reason referee #2 completely disregards that our protocol is based on the use of imaging flow-cytometry and NOT on conventional flow-cytometry, which has indeed been used in all mentioned papers. Indeed, in our experiments, we found that the combination of PLA with conventional FACS is very inaccurate and rather deceptive, as it cannot distinguish between true and false fluorescent signals. Importantly, we stress this specific problem throughout our manuscript (e.g. Fig. 2e,f, Supp.Fig3,4) and show that both high background signal arising from non-specific binding of the PLA probes (resulting in high false positive signal) or autofluorescence, cannot be distinguished by conventional flow-cytometry, but can be differentiated and resolved (gated out according to signal shape and localization within each cell) using our protocol of imaging flow cytometry. Therefore, this specific referee either completely missed the most critical point of our protocol, or has a personal interest to preclude its publication.

In addition, all previous publications utilizing conventional FACS in combination with PLA used primary antibodies directly conjugated to the PLA probes. This is a great technical limitation, which requires that each lab will prepare expensive custom-made antibodies for each target, making the method very costly and unattractive. Importantly, it has been shown that even this step is not sufficient to get rid off false positive signal due to non-specific binding of the conjugated antibodies (as referred to in the Andersen SS, et al paper from 2013). In contrast, our PLIC protocol overcomes this issue by the imaging analysis of signal shape and localization (filtering out autofluorescence and non-specific binding). This is also why in our protocol no PLA probe conjugation step is essential, allowing a cheaper and universal protocol without the need for expensive customization per target.

Furthermore, the previous studies using conventional FACS-PLA protocol were done either on cell lines or on abundant primary populations isolated from human blood. Therefore, we would like to stress that our PLIC protocol is the first protocol that provides a sensitive, quantitative and reproducible detection of PPI and PTM in

a) extremely rare populations and/or

b) populations defined by many surface markers (not just one, as done by the previous papers);

c) all in a physiological context (i.e no use of reporter/tagged transgenic mice).

Considering all the above points, we felt it was not desirable to directly attack and highlight the negatives of the previous FACS-PLA publications that referee #2 mentions, but rather to highlight the advantages of our protocol vis-à-vis other currently available systems (including conventional FACS-PLA). However, if advised by the editors, we can certainly refer to these manuscripts (e.g.

Leuchowius et al. 2009, Lof et al 2016, Andersen et al 2013) and further detail their limitations and inaccuracy and better explain how our protocol overcomes these and other technical issues.

2. The biological data presented in the manuscript are, as pointed out by the authors, not novel either. The only modest new data is the demonstration of “homo-oligomerization under physiological conditions. As all the technology and data presented here are previously known, my recommendation is to not try to publish this manuscript.

We are afraid this point mirrors the bad spirit of the previous one and is either based on ignorance of this referee or his/hers intention to fail our manuscript.

Obviously, to demonstrate the validity and reliability of our protocol, one has to first confirm a known interaction, which can then serve as a proof of concept. To this end, we validated interaction between Aire and Sirt1, which we previously reported to interact with each other under in vitro and in vivo conditions (Chuprin Nat Immunol 2015). Except for this specific result, our manuscript provides novel in-vivo data, which were never reported before, as no currently available methodology could offer a technically feasible protocol. Moreover, these data are quantitative, as opposed to ambiguous/inaccurate measurements obtained from artificial in-vitro system (e.g. transfected cell lines) which are neither quantitative nor they have a physiological significance.

Reviewers' comments:

Reviewer #1 (Remarks to the Author):

I would be happy to read the revised manuscript.

Reviewer #3 (Remarks to the Author):

In this manuscript, the authors describe a development of PLIC, a novel proteomic assay that enables a quantitative, robust and accurate analysis of PPIs and PTMs at a single cell level.

As a proof of principle, the authors used mTECs as an example of a rare immune population. By applying PLIC, the authors were able to obtain several novel in vivo biological findings, which provide important molecular insights into the mechanisms underlying the establishment of central immunological tolerance in the thymus.

In particular, the authors show for the first time that Aire undergoes homo-oligomerization under physiological conditions in a large fraction of mTECs that were directly isolated from a mouse thymus. To me, this finding is of particular importance as it helps better elucidating and validating the pathogenesis of recently identified mono-allelic mutations in the AIRE gene, which were found to cause common organ-specific autoimmune diseases in a dominant negative manner.

I think this manuscript is suited for publication in Nature Communications after the authors have addressed a couple of minor comments that are listed in the following:

(i) it would be great if the authors could compare their newly described PLIC technique with confocal microscopy and other imaging technologies

(ii) the authors should also comment on the drawback of the PLIC assay - they only discussed its advantages but no disadvantages were mentioned

Point-by-point response to the referees' comments on the manuscript by

Avin A. et al. (NCOMMS-17-11256)

We would like to thank the reviewers for their very helpful and valuable comments and suggestions, which enabled us to greatly improve the quality of our manuscript. We have accepted virtually all their comments/suggestions and tried to address each of them as thoroughly as possible. Our detailed point-by-point response (*highlighted in blue font*) to the specific comments is outlined below.

Revisions in the MS itself are also *highlighted in blue font*.

Referee #1

1. The expressions of CD80 in some samples look odd, Some explanations are needed. Without PLA treatment, do they find similar pattern of CD80 expression?

*Re: Indeed, the CD80 staining in some analyzed mTECs looks quite atypical in comparison to other surface molecules such as MHCII or EpCAM, which usually (*but not always*) give very clear signal that shows an increased intensity at the cells' surface. The reason for such a rather odd staining of CD80 is unclear to us. It may possibly reflect the quality of the commercially available antibody or changes of CD80 protein distribution on mTECs surface upon their fixation/permeabilization.*

We would like to stress, however, that even this staining is sufficient to easily distinguish between the mTEC^{high} (higher signal intensity) and mTEC^{lo} (no-low signal intensity) populations and correlates very well with MHC-II staining, which can be used as an additional/alternative mTEC maturation marker. Unfortunately, we could not use MHC-II staining in our experiments (and had to use CD80 staining instead), as the strain of Sirt1 KO mice that was used in our analyses is maintained on a mixed genetic background.

Importantly, CD80 staining looks the same whether followed by PLA or not, as can be seen in these images acquired in a separate experiment of primary antibody calibrations (without PLA). In both cases the cells are fixed and permeabilized in order to allow staining for nuclear proteins.

2. One of the strengths of this combined technique is to analyze hundreds or even thousands of primary cells in a mixed cell population. It is not clear whether the given quantitative data showed the results of this many cells. The figures mostly showed an average of 3 experiments instead of an average of 1,000 cells. More analytical data can be given if the cells show different patterns of Aire oligomerization and cell phenotype (including CD80 expression).

Re: We thank the reviewer for pointing out that the methods and figure legends do not sufficiently explain to the reader how our data were acquired and analyzed. We have modified the relevant text and now detail the experimental outline and analysis strategy in more depth and clarity, both in the methods and figure legends.

Specifically, each analysis is based on statistical evaluation of 3 independent experiments. Each independent experiment is based on analysis of 2-4 biological replicates. Each biological replicate consists of Imaging flow cytometry analysis of approximately 1,000 TECs (CD45-EpCAM+) and control cells (CD45+EpCAM-) or of approximately 20,000 BMDM (F4/80+CD11b+) for inflammasome experiments.

Determination of true PLA signal is explained in detail in the method section, but basically correlates to signal intensity exceeding the probes only sample threshold. It is important to stress that we quantify the percentage of PLA+ cells, rather than the intensity of PLA signal per cell, which is not a quantitative measure of protein proximity. The reason for this is that while the PLA signal can look differently between cells (intensity and area wise), it doesn't necessarily correlates to more interactions and/or to different interactions pattern. It could arise from technical differences between samples (i.e. antibody binding efficiency, probes concentration per cell, polymerase efficiency in creating the rolling circle amplification). Based on our experience, assessment of % of positive signal within the cell population serves as the most accurate and reliable analysis, especially if acquired in >3 independent experiments each consisting of >2 biological replicates. We now address this point and elaborate on it in detail in the discussion. We have also made adequate modifications in the methods and figure legends to further clarify this point.

3. It will be more informative if the authors could compare their technique with confocal microscopy and other imaging technology.

Re: Indeed, this is a great point and we would like to thank referee #1 for suggesting it.

In the revised manuscript we have included new data and figures, which better highlight the advantages of PLIC in comparison to conventional PLA linked to confocal microscopy (Fig. 1, Supp. Fig 1b,2) and/or PLA coupled to flow cytometry (Supp. Fig 3).

Specifically, these data highlight that in analyses of tissue sections that is done by conventional PLA followed by confocal microscopy, the signal cannot be accurately quantified and linked to a specific cell population, and that tissue sections can be extremely heterogeneous (Supp. Fig 2), leading to over- or under-representation of a specific cell population in a given tissue section, especially when such a population is rare. Similarly, in Fig 2, we demonstrate that in contrast to PLIC, PLA linked to conventional flow cytometry is unable to effectively discriminate between true and false positive PLA signal and may consequently provide highly inaccurate data.

Correspondingly, we have substantially modified and expanded the discussion, in which we now detail the advantages and disadvantages of PLIC in comparison with the currently available

methods for assessing proteins proximity including conventional PLA, FRET, PCA, BioID, APEX, etc.

Referee #3

1. It would be great if the authors could compare their newly described PLIC technique with confocal microscopy and other imaging technologies

Re: Indeed, this is a great point, also highlighted by referee #1. As already mentioned above, we have included new data and figures, which better highlight the advantages of PLIC in comparison to conventional PLA linked to confocal microscopy (Fig 1, Supp. Fig 1b, 2) and/or PLA coupled to flow cytometry (Supp. Fig 3).

We have also addressed and further elaborated on the advantages that PLIC offers in comparison to classical fluorescence microscopy (see above response to referee #1) and other proximity based imaging methods such as PLA, FRET, PCA, BioID, APEX, etc. in the discussion section.

2. The authors should also comment on the drawback of the PLIC assay - they only discussed its advantages but no disadvantages were mentioned

Re: We completely agree with referee #3 on this point. We have now added a paragraph in the discussion section addressing potential PLIC limitations and suggest possible solutions to these limitations. We have also compared between PLIC and other imaging technologies, or other PLA based methods, in order to better portray the advantages and disadvantages of our new PLIC protocol compared to the existing methods.

With very best wishes,

Jakub Abramson

REVIEWERS' COMMENTS:

Reviewer #1 (Remarks to the Author):

The authors have addressed the questions raised by reviewers. The revised manuscript is now suited for publication in Nature Communications.

Reviewer #3 (Remarks to the Author):

The authors have addressed all points that I raised in my primary review.